# WMPO: World Model-based Policy Optimization for Vision-Language-Action Models

**Fangqi Zhu[1], Zhengyang Yan[1], Zicong Hong[1], Quanxin Shou[1], Xiao Ma[2†], Song Guo[1†]**
[1]Hong Kong University of Science and Technology [2]ByteDance Seed
fangqi.zhu@connect.ust.hk, xiao.ma@bytedance.com, songguo@ust.hk
Project Page: https://wm-po.github.io/

## Abstract

Vision-Language-Action (VLA) models have shown strong potential for general-purpose robotic manipulation, but their reliance on expert demonstrations limits their ability to learn from failures and perform self-corrections. Reinforcement learning (RL) addresses these through self-improving interactions with the physical environment, but suffers from high sample complexity on real robots. We introduce *World-Model-based Policy Optimization (WMPO)*, a principled framework for on-policy VLA RL without interacting with the real environment. In contrast to widely used latent world models, WMPO focuses on pixel-based predictions that align the *"imagined"* trajectories with the VLA features pretrained with web-scale images. Crucially, WMPO enables the policy to perform on-policy GRPO that provides stronger performance than the often-used off-policy methods. Extensive experiments in both simulation and real-robot settings demonstrate that WMPO (i) substantially improves sample efficiency, (ii) achieves stronger overall performance, (iii) exhibits emergent behaviors such as self-correction, and (iv) demonstrates robust generalization and lifelong learning capabilities.

## 1 Introduction

Vision-Language-Action (VLA) models (Brohan et al., 2023; Kim et al., 2024; Intelligence et al., 2025) have emerged as a promising paradigm for general-purpose robotic manipulation, enabling robots to follow natural language instructions in complex, unstructured environments. The dominant approach for training these models is imitation learning (IL) from large-scale human demonstrations (Lin et al., 2024). While effective in mimicking demonstrated behaviors, IL-trained policies are often brittle. When encountering out-of-distribution states not seen during training, they can take suboptimal actions that lead to compounding errors (Ross et al., 2011), making task completion or recovery nearly impossible (Fig. 1a).

Reinforcement learning (RL) (Li et al., 2025b) offers a natural solution to this brittleness by allowing an agent to learn and improve through active interaction with its environment. This self-improvement process can lead to policies that are more robust and capable of recovering from failure. However, applying RL directly to real robots is notoriously sample-inefficient, requiring millions of interactions that are impractical, unsafe, and time-consuming to collect (Fig.1b). Recent efforts to improve sample efficiency fall into two main strategies. The first leverages human intervention to guide learning (Luo et al., 2024b; Chen et al., 2025; Xia et al., 2025), which reduces exploration cost but is labor-intensive and hard to scale. The second relies on simulation to reduce real-world interactions (Lu et al., 2025; Li et al., 2025a), but is limited by the difficulty of building accurate simulators for diverse scenarios.

The advent of large-scale generative models, particularly video-generative world models (NVIDIA et al., 2025; Ball et al., 2025), presents a compelling new frontier for leveraging model-based RL (Finn & Levine, 2017) to mitigate the sample inefficiency challenge in VLA RL. These models can learn the dynamics of the world from data and simulate future transitions, offering a path to scalable RL without costly real-world explorations. Nevertheless, integrating these models with existing VLAs remains a challenge. Many classical model-based RL approaches (Hafner et al., 2019; 2020a;b;

---

†Corresponding authors.

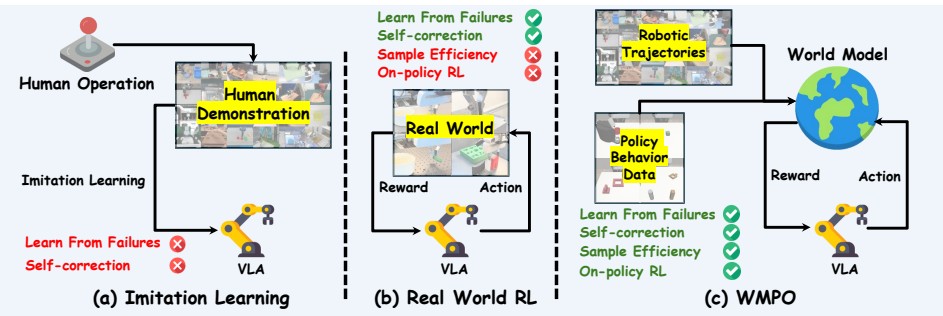

Figure 1: Three different VLA training paradigms: (a) Imitation learning learns from human demonstrations but lacks the ability for learning from failures and self-correction; (b) Real-world RL improves policy through direct interaction but suffers from high sampling costs and difficulty in achieving on-policy RL; (c) WMPO pretrains a world model on large-scale robotic trajectories and fine-tunes it with limited policy behavior data, enabling sample-efficient on-policy RL for VLA without real-world interaction.

2023; Ma et al., 2022; Frauenknecht et al., 2025) operate in an abstract latent space, which creates a fundamental mismatch with powerful VLA foundation models that are pretrained on real-world images. The rich, pretrained visual understanding of VLAs cannot be directly applied within a mismatched latent dynamics model. We argue that leveraging a pixel-space video-generative world model is crucial, as it allows the VLA policy to operate on generated visual data that is consistent with its pretraining, effectively bridging the world model with the policy's pre-trained knowledge.

To this end, we propose World Model-based Policy Optimization (WMPO), as illustrated in Fig. 1c, a principled framework that grounds VLA RL entirely in an action-conditioned video world model. By pretraining a high-fidelity, pixel-space video-generative model on millions of robotic trajectories(Collaboration, 2023), WMPO leverages realistic visual dynamics to create a scalable training environment for downstream tasks.

WMPO incorporates several key innovations. First, to mitigate the state-distribution mismatch between expert demonstrations and policy rollouts, we introduce policy behavior alignment, finetuning the world model with behavioral data collected by the policy itself. Second, short-horizon prediction makes it difficult to define accurate rewards and is prone to reward hacking. To address this, WMPO generates complete trials through clip-level autoregressive video generation, enabling more reliable outcome-based reward assignment. While this design supports long-horizon rollouts, it also introduces challenges such as visual distortion and action–frame misalignment. WMPO addresses these with noisy-frame conditioning and frame-level action control, ensuring robustness and accurate trajectory simulation. Third, we train a lightweight reward model that predicts task success or failure, providing a learned sparse reward signal and avoiding both complex reward shaping and reward hacking. Together, these components form a self-contained environment that enables on-policy RL entirely "in imagination". Specifically, we adopt Group Relative Policy Optimization (GRPO) (Shao et al., 2024), whose robustness and scalability have been demonstrated in DeepSeek-AI et al. (2025). Notably, WMPO naturally supports repeated rollouts from the same initial state, which is difficult to realize in the physical world but crucial for large-scale GRPO training.

We conduct extensive experiments in both Mimicgen simulation environments (Mandlekar et al., 2023) and real-world environments to validate the effectiveness of WMPO. Our results show that WMPO achieves substantially higher sample efficiency and consistently outperforms VLA RL methods that directly optimize with real trajectories. Crucially, we provide qualitative evidence of emergent behaviors, where the WMPO-trained policy demonstrates self-correction strategies not present in the demonstration data and often completes tasks faster and more smoothly, without noticeable stalls. We further demonstrate WMPO's strong generalization compared to offline RL methods, as well as its capacity for lifelong learning through alternating updates between the VLA policy and the world model. Taken together, these findings highlight WMPO as a scalable and generalizable paradigm for advancing VLA RL.

## 2 RELATED WORK

**Vision-Language-Action Models.** Vision-Language-Action (VLA) models aim to map visual inputs and natural language instructions into executable robotic actions, enabling general-purpose manipulation. Most VLAs build upon pretrained vision-language models (VLMs) and are further fine-tuned on robotic trajectories, thereby inheriting strong visual grounding and language understanding. This progress has been driven both by large-scale demonstration collections (Collaboration, 2023) and by advances in action parameterization, including discrete next-token prediction (Kim et al., 2024), parallel decoding (Kim et al., 2025), and continuous flow-based heads (Black et al., 2024). Despite these advances in data and model design, existing VLAs largely remain within the imitation learning (IL) paradigm, making them brittle when encountering out-of-distribution states and unable to leverage failed executions for improvement (Ross et al., 2011).

**Reinforcement Learning for VLA Models.** RL provides a natural complement to IL by enabling policies to learn from interaction, thereby improving robustness and recovery capabilities. However, applying on-policy RL to VLA remains challenging due to severe sample inefficiency and substantial system-level complexity. Prior works can be broadly divided into two strategies. The first introduces human intervention to guide exploration, e.g., supplying corrective signals when policies encounter unrecoverable states (Luo et al., 2024a; Chen et al., 2025; Xia et al., 2025). While effective at reducing exploration cost, this approach requires continuous human supervision, making it labor-intensive and difficult to scale. The second attempts to perform RL directly in simulation or in the real world. For instance, Zhang et al. (2024) adopt trajectory-level DPO (Rafailov et al., 2023), while others apply PPO (Schulman et al., 2017) or GRPO (Shao et al., 2024) to optimize VLA policies in simulation (Lu et al., 2025; Li et al., 2025a). These approaches avoid human supervision but still suffer from poor sample efficiency, and constructing accurate simulators for each real-world scenario introduces prohibitive engineering overhead. In contrast, WMPO enables policy optimization entirely within a learned world model, substantially improving sample efficiency and scalability.

**World Models.** World models aim to mitigate the need for costly real-world interactions by learning generative dynamics and enabling policies to learn from "imagine" trajectories. Early approaches (Hafner et al., 2020a;b; 2023) learned world model in the latent space of recurrent state-space models, which provided efficient but overly abstract rollouts. More recent work introduced diffusion-based world models, showing that retaining pixel-level fidelity is crucial for RL with Gaussian policies (Alonso et al., 2024; Jiang et al., 2025). Building on this trend, large-scale video world models (NVIDIA et al., 2025; Ball et al., 2025) have demonstrated impressive generality across diverse domains. However, when applied to robotics, they suffer from distribution mismatch—struggling to faithfully reproduce policy rollouts and fine-grained robot–object interactions. In contrast, WMPO addresses these challenges through policy behavior alignment and, for the first time, verifies the feasibility of leveraging high-fidelity world models for scalable VLA RL.

## 3 WORLD MODEL-BASED POLICY OPTIMIZATION

We introduce World Model-based Policy Optimization (WMPO), a novel framework for learning complex, vision-based robotic manipulation policies. The WMPO framework is grounded in the principles of model-based RL, where policy optimization is performed entirely within a learned world model, thereby eliminating the need for costly real-world interactions. WMPO operates directly in the pixel space, instead of latent space, which better bridges the pretrained VLA features from web-scale video-action pairs with the "imagined" trajectories. Specifically, the world model is trained to accurately reflect the policy's behavior through a process we call Policy Behavior Alignment, where it is fine-tuned on a small set of real trajectories collected from the policy itself. This ensures the model can faithfully simulate diverse scenarios, including failures. We also introduce noisy frame conditioning and frame-level action control techniques to overcome the problems of visual distortion and action-frame misalignment in long-horizon video prediction. With these modifications, we are able to perform strong on-policy Group Relative Policy Optimization (GRPO) using trajectories imagined by the learned world model, which significantly enhances sample efficiency compared to direct RL methods. Fig. 2 illustrates an overview of the training procedure of WMPO.

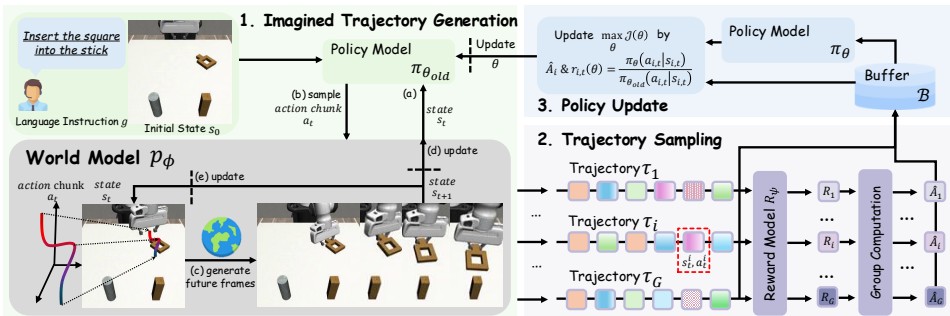

Figure 2: WMPO starts from an initial state $s_0$. The overall training procedure consists of three components: (1) *Imagined Trajectory Generation*, where policy model $\pi_{\theta_{old}}$ and world model $p_\phi$ interact alternately to generate a full imagined trajectory; (2) *Trajectory Sampling*, where multiple trajectories are sampled and evaluated by the reward model $R_\psi$; and (3) *Policy Update*, where the policy parameters $\theta$ are optimized via Eq. 4. This process is iteratively repeated throughout training.

## 3.1 PROBLEM FORMULATION

We formulate the VLA manipulation task as a decision-making problem in the form of an MDP $\mathcal{M} = (\mathcal{S}, \mathcal{A}, P, R)$.

- **State space.** $\mathcal{S} = \mathcal{I} \times \mathcal{G}$, where $\mathcal{I}$ denotes the image observation space, i.e., image sequences $I_{0:K}$, and $\mathcal{G}$ denotes the language instruction space. Here, we make the assumption that robot states can be solely defined by their image observations. For more complicated setups, e.g., Partially Observable MDPs (POMDPs) (Igl et al., 2018; Ma et al., 2021), we leave them for future studies.

- **Action space.** $\mathcal{A}$ denotes the space of action chunks, i.e., sequences of robot actions. Each action in an action chunk of length $K$, $a_t \in \mathbb{R}^{K \times D}$, represents a $D$-DoF (degree of freedom) control vector. For policy optimization, each dimension is discretized into 256 bins. Specifically, we learn to sample action chunks using a parameterized VLA policy, $a \sim \pi_\theta(a \mid s)$.

- **Transition function.** $P : \mathcal{S} \times \mathcal{A} \rightarrow \mathcal{S}$ is realized by a parameterized world model $s_{t+1} \sim p_\phi(s_{t+1} \mid s_t, a_t)$, which generates future observations conditioned on past observations and actions. In particular, we sample *imagined trajectories* $\tau = \{s_0, \hat{s}_1, \ldots, \hat{s}_T\}$ by iteratively sampling from the world model $\hat{s}_{t+1} \sim p_\phi(s_{t+1} \mid \hat{s}_t, a_t)$ and the VLA policy $a_t \sim p_\theta(a_t \mid s_t)$, given an initial state $s_0$ sampled in the real environment.

- **Reward function.** The reward is given by a learned model $R_\psi$ that determines task success from the full trajectory, assigning a binary outcome $R_\psi(\tau) \in \{0, 1\}$.

Our objective is to train a policy $\pi_\theta(a \mid s)$ such that the predicted cumulative return of the imagined trajectories will be maximized

$$\max_\theta \mathbb{E}_{\tau \sim \pi_\theta, p_\phi} \left[ R_\psi(\tau) \right]. \tag{1}$$

This formulation highlights a general paradigm: RL for VLA can be decoupled from real-world interactions by leveraging a generative world model as the imaginary training ground.

## 3.2 GENERATIVE WORLD MODEL

**Imagined Trajectory Generation.** Given $c$ initial frames $I_{0:c}$, the policy $\pi_\theta$ takes the most recent $m$ frames and language instruction $g$ as input and predicts an action chunk [1] , i.e., $a_{i:i+K} \sim \pi_\theta(I_{i-m:i}, g)$. The world model $p_\phi$ then conditions on the last $c$ observed frames and the predicted action chunk to generate the next $K$ frames:

$$I_{i:i+K} \sim p_\phi(I_{i-c:i}, a_{i:i+K}). \tag{2}$$

---

[1]To avoid confusion in subscripts, we specify that $i$ is used as the frame-level subscript and $t$ as the state-level subscript, where a state often includes multiple frames. $N$ and $T$ denote the maximum length of an imagined trajectory at the frame level and state level, respectively.

Repeating this process until a maximum length $N$ yields a trajectory $\tau = \{I_{0:N}, a_{0:N}\}$. A reward model $R_\psi$ evaluates the frames in $\tau$ and outputs a binary label $y = R_\psi(I_{0:N}) \in \{0, 1\}$. Thus, each imagined trajectory in the world model is represented as a labeled pair $(\tau, y)$, which is then used for policy optimization.

**Model Architecture.** Our world model is based on a video diffusion backbone inherited from OpenSora (Zheng et al., 2024), with modifications designed for simulating robot–object interactions. Specifically, we replace the 3D VAE in OpenSora with the 2D VAE from SDXL (Podell et al., 2024), which better preserves fine-grained motion details and avoids temporal distortions caused by excessive compression. Consequently, the diffusion process operates in the VAE's latent space. When applying the imagined trajectory to VLA optimization, we decode the images back into pixel space to better leverage the pretrained knowledge, rather than retraining the VLA in a new latent space such as that from RSSM (Hafner et al., 2023).

Since our world model generates full trajectories in an autoregressive manner—using previously generated frames as conditioning for future predictions—errors can accumulate, leading to severe degradation over long horizons, ultimately leading to failed prediction. To mitigate this issue, we introduce a noisy-frame conditioning technique: during training, conditional frames $I_{i-m:i}$ are perturbed with mild diffusion noise corresponding to an early timestep (i.e., the noise level at step 50, where step 1000 would correspond to pure Gaussian noise), rather than kept clean. This improves robustness to imperfect conditioning.

As a result, the world model achieves stable long-horizon generation, producing trajectories of hundreds of frames without noticeable quality loss.

To enable precise action conditioning, inspired by Zhu et al. (2025), we incorporate a frame-level action control mechanism for better action-frame alignment. Specifically, we extend the AdaLN (Xu et al., 2019) block to inject both action signals and diffusion timestep embeddings at the frame level. For each action $a_i$, an MLP generates modulation coefficients: scale $\gamma_1^i$ and shift $\beta_1^i$ for the LayerNorm output, and scale $\alpha_1^i$ for the residual connection of either the MHA or FFN block. Let $\mathbf{x}^i$ denote the feature representation at frame $i$; the update rule within each transformer block is given as:

$$\mathbf{x}^i = \mathbf{x}^i + (1 + \alpha_1^i) \cdot \text{Block}\Big(\gamma_1^i \cdot \text{LayerNorm}(\mathbf{x}^i) + \beta_1^i\Big).$$

**Policy Behavior Alignment.** We pretrain the world model on Open X-Embodiment (OXE) Collaboration (2023) trajectories, which offer diverse demonstrations of robot interactions and endow the model with broad knowledge of physical dynamics. However, because OXE trajectories primarily consist of successful executions, failure scenarios are underrepresented in the observation distribution. Likewise, training only on expert demonstrations from downstream tasks leaves the model unable to simulate failures, making the imagined trajectories unsuitable for training. To address this mismatch, we fine-tune the world model on real rollout trajectories collected from the policy itself, thereby adapting it to the downstream (state, action) distribution and capturing failure modes more faithfully. Without this adaptation, the model's imagination of failure cases remains brittle and unfaithful.

### 3.3 REWARD MODEL

A key requirement for scalable policy optimization in the world model is automatically judging whether an imagined trajectory indicates task success. We construct a lightweight reward model trained on real trajectories. Given a trajectory $\tau = \{I_{0:N}\}$, we define a clip of length $L$ as $c_i = I_{i-L:i}$. The terminal clip $c_N$ of a successful trajectory serves as a positive sample, whereas negatives are drawn from $\{c_i : L \leq i \leq N - L\}$ of successful trajectories and from arbitrary clips of failed trajectories. To address class imbalance, we balance the number of positive and negative samples within each training batch. The reward model, implemented as a VideoMAE (Tong et al., 2022) encoder with a linear head, is trained with binary cross-entropy loss. At inference, the model applies a sliding window with stride $s$ over $\tau$ to compute the success probability of each clip. A trajectory is classified as successful if any clip exceeds a threshold $\tau_{\text{thr}}$, which is selected via validation experiments.

### 3.4 ON-POLICY REINFORCEMENT LEARNING WITH WMPO

Reinforcement learning in VLA tasks faces two key bottlenecks. (1) Physical interaction bottleneck. Unlike LLM settings where feedback is cheap, VLA tasks require repeated rollouts in the physical world, which incur high hardware costs, safety concerns, and limited scalability. (2) Value estimation bias by off-policy RL. Given the aforementioned physical constraints, existing real-world RL methods often consider off-policy RL methods (James et al., 2022; Seo et al., 2024; Wagenmaker et al., 2025; Chen et al., 2025). However, off-policy methods naturally cause value estimation errors (Park et al., 2025) and on-policy methods are often favorable for better performance.

To overcome these challenges, we optimize policies entirely within a world model: replacing costly real-world rollouts with model-generated trajectories alleviates reliance on physical interaction and enables scalable online learning. We adopt Group Relative Policy Optimization (GRPO) as the policy optimization algorithm, since it provides stable and scalable training in settings with sparse rewards. In our case, state transitions are simulated by the world model (Eq. 2), and rewards are binary with $R_\psi(\tau) \in \{0, 1\}$ depending on task success.

**Trajectory Sampling.** From each initial frames $I_{0:c}$ sample from real environment $\mathcal{D}$, we sample a group of imagined trajectories $\{\tau_1, \ldots, \tau_G\}$ from current policy $\pi_{\theta_{old}}$ inside the world model. The reward model is then employed to predict the success or failure of each trajectory. To mitigate vanishing gradients, we adopt a *Dynamic Sampling* strategy (Yu et al., 2025): if all trajectories in a group are predicted to be successful or all unsuccessful, the group is discarded and additional rollouts are sampled until the batch is fully populated. The log-probability of each action chunk under $\pi_{\theta_{old}}$ is pre-computed as reference:

$$\log \pi_{\theta_{old}}(a_t \mid s_t) = \sum_{i=1}^{K} \sum_{j=1}^{D} \log \pi_{\theta_{old}}\left(a_t^{i,j} \mid s_t\right), \tag{3}$$

where $a_t$ denotes the action chunk at time $t$, and $a_t^{i,j}$ represents the $i$-th action in the $j$-th DoF.

**Policy Update.** Following DAPO (Yu et al., 2025), we remove the KL divergence regularization so that no reference model is required during training, thereby reducing memory consumption and encouraging the policy to explore novel behaviors. The final training objective is given by

$$\mathcal{J}(\theta) = \mathbb{E}_{s_0 \sim \mathcal{D}, \{\tau_i\}_{i=1}^{G} \sim \pi_{\theta_{old}}} \left[ \frac{1}{G} \sum_{i=1}^{G} \frac{1}{T} \sum_{t=0}^{T} \min\left(r_{i,t}(\theta)\hat{A}_i, \ \mathrm{clip}(r_{i,t}(\theta), 1 - \epsilon_{low}, 1 + \epsilon_{high})\hat{A}_i\right) \right], \tag{4}$$

with

$$r_{i,t}(\theta) = \frac{\pi_\theta(a_{i,t} \mid s_{i,t})}{\pi_{\theta_{old}}(a_{i,t} \mid s_{i,t})}, \qquad \hat{A}_i = \frac{R_i - \mathrm{mean}(\{R_i\}_{i=1}^{G})}{\mathrm{std}(\{R_i\}_{i=1}^{G})}. \tag{5}$$

Here $r_{i,t}(\theta)$ is the probability ratio between new and old policies at step $t$ of trajectory $\tau_i$, $R_i = R(\tau_i)$, and $\hat{A}_i$ is the normalized advantage of trajectory $\tau_i$ over the horizon $N$. The overall training pipeline is detailed in Algorithm 1 in Appendix A.

## 4 EXPERIMENTS

We conduct extensive experiments to evaluate the effectiveness of WMPO, focusing on the following questions: (1) can WMPO outperform online and offline RL in simulation environments; (2) how does the behavior of WMPO differ from imitation learning; (3) can WMPO generalize to unseen settings; (4) can WMPO achieve iterative improvement during deployment; and (5) can WMPO be applied on a real robot?

### 4.1 EXPERIMENT SETTINGS

**Implementation Details** In this work, we fine-tune OpenVLA-OFT (Kim et al., 2025) via imitation learning on target manipulation tasks as our base policy. For simplicity, we omit the robot proprioceptive state and wrist camera inputs, and set the action chunk length $K$ to 8. We collect $P$ real

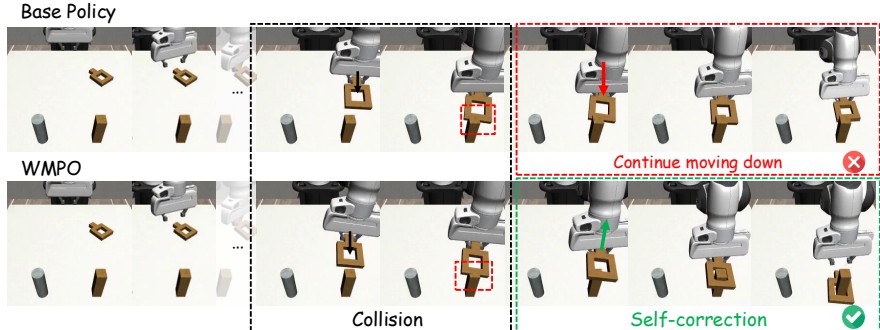

Figure 3: Behavior analysis of the *Square* task (inserting the square into the stick) shows that, compared with the base policy, WMPO demonstrates the ability to self-correct.

trajectories using the base policy, and conduct experiments with $P = 128$ and $P = 1280$ to evaluate the scalability of our approach. These trajectories are further used to fine-tune a world model, which predicts the next $K = 8$ frames given $c = 4$ conditioning frames and one action chunk. In addition, we train reward models on the collected trajectories, using a video clip length of $L = 8$ and a stride of 1 during evaluation. Further training details are provided in Appendix A.

**Simulation Environment Details**   We conduct experiments in the Mimicgen simulation (Mandlekar et al., 2023). We select four fine-grained manipulation tasks: *Coffee_D0*, *StackThree_D0*, *ThreePieceAssembly_D0*, and *Square_D0*, and fine-tune the OpenVLA-OFT model with 300 expert trajectories per task as the base policy. For evaluation, we test 128 different initial states for each task and report the average success rate.

## 4.2   COMPARISON EXPERIMENTS

We compare WMPO with two established RL algorithms, GRPO (Shao et al., 2024) and DPO (Rafailov et al., 2023), both widely used for optimizing large language models. To ensure fairness, all methods are allocated the same real rollout budget $P$. We consider both online and offline baselines: GRPO is implemented in an online setting, where the policy is updated directly from trajectories collected in the environment; DPO is implemented in an offline setting, where the base policy serves as the reference and trajectory pairs (success vs. failure) are constructed for optimization using the standard DPO loss. Unlike GRPO, which discards trajectories after each update, DPO can repeatedly reuse collected data, but it lacks the ability to update the policy online as WMPO does. More implementation details are provided in Appendix B.

As shown in Tab. 1, WMPO consistently outperforms all baselines across all tasks. With a small rollout budget of $P$=128, it already surpasses the strongest baseline by +9.8 points, demonstrating

Table 1: Comparison of policy optimization methods across four manipulation tasks in the Mimicgen simulation benchmark. $P$ denotes the rollout budget, i.e., the number of full real trajectories available for policy optimization. Results show that WMPO consistently outperforms both GRPO and DPO baselines under different budgets. As the rollout budget increases from 128 to 1280, WMPO continues to exhibit substantial improvements, highlighting both its data efficiency and scalability. Performance is reported as the task success rate (%).

| Rollout budget $P$ | Methods | Coffee | StackThree | ThreePieceAssembly | Square | Mean (%) |
|---|---|---|---|---|---|---|
| – | *Base policy* | 43.8 | 46.9 | 19.5 | 24.2 | 33.6 |
| 128 | GRPO | 38.3 | 52.3 | 17.2 | 25.0 | 33.2 |
| | DPO | 43.8 | 53.9 | 23.4 | 28.1 | 37.3 |
| | Ours | **61.7** | **56.3** | **37.5** | **32.8** | **47.1** |
| 1280 | GRPO | 47.7 | 54.7 | 20.3 | 25.8 | 37.1 |
| | DPO | 52.3 | 57.0 | 26.7 | 33.6 | 42.4 |
| | Ours | **75.0** | **64.1** | **46.1** | **45.3** | **57.6** |

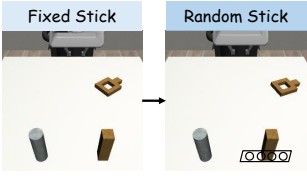 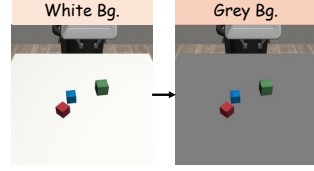 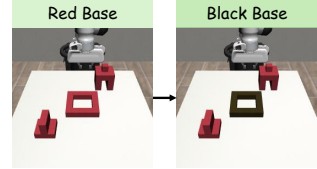

(a) Position Disruption          (b) Background Disruption          (c) Texture Disruption

Figure 4: (a) For the *Square* task, we vary the stick's position from fixed to a random position inside a rectangle. (b) For the *StackThree* task, we substitute the tabletop background with a gray background. (c) For the *ThreePieceAssembly* task, we substitute the red base with a dark wooden base.

strong data efficiency. When the budget increases to $P=1280$, the margin further expands to $+15.2$ points on average, indicating that WMPO leverages additional trajectories more effectively than existing methods and scales robustly with more rollouts. In contrast, GRPO often underperforms with limited updates, and DPO plateaus due to static data reuse, whereas WMPO continues to improve steadily as the rollout budget grows. Furthermore, we evaluate the reward model and find that it achieves an F1 score above 0.95 across all tasks, reliably distinguishing success from failure and effectively mitigating reward hacking. These results highlight the effectiveness and scalability of WMPO for policy optimization in robotic manipulation.

### 4.3 EMERGENT BEHAVIOR OF WMPO

To better understand the source of WMPO's strong performance, we conduct a visual comparison of its test-time behavior against the base policy. We identify two emergent behaviors unique to WMPO: (1) the WMPO policy learns to self-correct, recovering from nearly failure states; and (2) the WMPO policy executes tasks more efficiently, as it rarely becomes "stuck" in suboptimal states.

First, taking the *Square* task as an illustrative example (see Fig. 3), we observe that when both the base policy and WMPO deviate from the correct trajectory due to error accumulation and encounter a collision, their behaviors diverge. The baseline policy, trained only on expert demonstrations, has never observed collisions during training; it continues to push the square against the stick until the maximum time horizon is reached, resulting in failure. In contrast, WMPO benefits from large-scale imagined trajectories generated by the world model, enabling it to learn the self-correction behaviors, which is challenging to obtain through imitation learning alone. Specifically, the policy autonomously learns to lift the square, realign it, and then insert it correctly, ultimately succeeding in the task. Second, we analyzed the lengths of successful trajectories generated by different policies, as shown in Fig. 5. We found that the trajectories of policies trained with WMPO are significantly shorter. This is because WMPO discourages stuck behaviors, which often result in failures due to timeouts. As a side benefit, WMPO also makes the policy's behavior faster and smoother.

### 4.4 GENERALIZATION TO NOVEL TASKS

In this section, we evaluate the generalization ability of WMPO across three novel disruption scenarios (Fig. 4), which systematically assess generalization under spatial, background, and texture shifts. As shown in Tab. 2, WMPO consistently achieves the best performance across all disruption types. DPO attains modest improvements in the in-distribution setting compared to the base policy, but its performance degrades significantly under background and texture changes, suggesting reliance on spurious visual cues rather than transferable manipulation skills. GRPO

Table 2: We evaluate each policy in its corresponding disruption scenario and report the success rate (%).

| Methods | Pos. Dis. | Bg. Dis. | Tex. Dis. | Mean |
|---|---|---|---|---|
| Base policy | 14.1 | 46.1 | 10.9 | 23.7 |
| GRPO | 15.6 | 47.7 | 10.9 | 24.7 |
| DPO | 16.4 | 34.4 | 7.8 | 19.5 |
| **Ours** | **22.3** | **50.0** | **16.4** | **29.6** |

exhibits performance similar to the base policy, and both are worse than WMPO across all disruption scenarios. In contrast, WMPO, trained entirely in the world model, captures more generalizable strategies and maintains reliable performance across spatial, background, and texture variations.

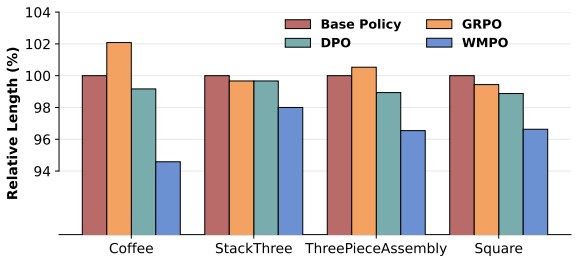

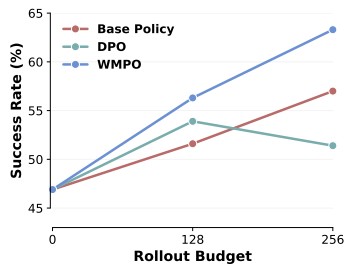

Figure 5: Relative average trajectory length of successful trials across different policies (Base Policy = 100%).

Figure 6: Lifelong learning results of WMPO and baselines.

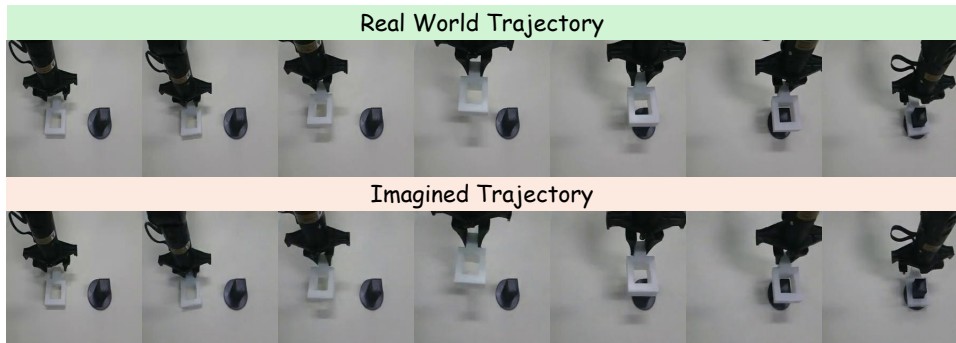

Figure 7: Real-world experiments on the fine-grained manipulation task "Insert the square into the stick" where the clearance between the square and the stick is only 5 mm. The top row shows the real-world trajectory of the base policy executed in the real world, while the bottom row depicts the corresponding imagined trajectory starting from the same initial state within our learned world model. Despite never observing this trajectory during training, the world model accurately predicts the future evolution, demonstrating its ability to capture precise task dynamics.

## 4.5 LIFELONG LEARNING

In this section, we demonstrate that WMPO can continuously improve the performance of VLA by iteratively collecting real trajectories from the environment. Specifically, we iteratively collect $P = 128$ real trajectories, perform WMPO to optimize the policy, and then use the updated policy to collect another $P$ real trajectories. We apply the same setting to the DPO baseline. To compare WMPO with an imitation learning-based policy using more expert demonstrations, we leverage 300, 428, and 556 expert trajectories to train the base policy as a reference. It is important to note that the base policy requires human-collected trajectories, whereas WMPO only relies on trajectories collected by the policy itself, making it more scalable. The results on the *StackThree* task, shown in Fig. 6, demonstrate that WMPO achieves stable and substantial improvements over both baselines, whereas DPO fails to improve iteratively due to unstable training.

## 4.6 REAL-WORLD EXPERIMENTS

In this section, we evaluate the challenging real-world manipulation task, "Insert the square into the stick" (see Fig. 7, more cases including failure could be found in Appendix C), to validate the effectiveness of WMPO. Using the Cobot Mobile ALOHA platform, we collect 200 high-quality expert demonstrations to fine-tune the OpenVLA-OFT model as the base policy. We then deploy this policy to collect an additional 128 trajectories, which are used to further fine-tune the world model and optimize the policy within it. For comparison, we also train an offline DPO policy using the same dataset. All models are evaluated under identical experimental conditions, and we report the average success rate over 30 trials. The results show that the base policy, DPO, and WMPO achieve success rates of 53%, 60%, and 70%, respectively, demonstrating the effectiveness of WMPO on real robots.

## 5 CONCLUSION

In this work, we introduced WMPO, a novel framework for on-policy RL of VLA models. By grounding policy optimization entirely in a video-generative world model, WMPO eliminates the need for costly real-world interactions while maintaining consistency with pretrained VLA representations. Through policy behavior alignment, robust autoregressive video generation, and a lightweight reward model, WMPO enables scalable training with strong sample efficiency. Extensive experiments in both simulation and real-world settings demonstrated that WMPO (i) consistently outperforms state-of-the-art model-free baselines, (ii) exhibits emergent self-correction behavior, (iii) generalizes reliably to unseen scenarios, and (iv) supports iterative lifelong learning. Together, these findings highlight WMPO as a scalable and generalizable paradigm for advancing VLA RL.

## 6 ETHICS STATEMENT

This work uses only publicly available, open-source datasets for training and evaluation. No private or sensitive data is involved. The proposed methods are intended solely for academic research in machine learning and robotics. No foreseeable negative societal impacts are anticipated beyond standard considerations for robotic learning research.

## 7 REPRODUCIBILITY STATEMENT

We have made every effort to ensure the reproducibility of our work. All datasets used in this study are publicly available and open source. Detailed descriptions of the experimental settings, hyperparameters, and training protocols are provided in Appendix A. We also release our implementation at https://github.com/WM-PO/WMPO, which includes training scripts, evaluation pipelines, and configuration files for all experiments. To further support downstream research, we additionally provide pretrained checkpoints of the world model and task-specific fine-tuned models, enabling researchers to directly reproduce our results and easily adapt the models to new tasks.

## ACKNOWLEDGMENTS

We are very grateful to Dr. Hongtao Wu for his insightful discussions and valuable suggestions in the early stage of this work. This research was supported by fundings from the Hong Kong RGC General Research Fund (152228/23E, 162161/24E, 162116/25E, 162180/25E), National Natural Science Foundation of China (NSFC) Key Program (No.62532005), Collaborative Research Fund (No. C1042-23GF, No. 5097-25G), NSFC/RGC Collaborative Research Scheme (Grant No. 62461160332 & CRS_HKUST602/24), Research Impact Fund (No. R5011-23F), Areas of Excellence Scheme (AoE/E-601/22-R), and the InnoHK (HKGAI).

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

# APPENDIX

## A   TRAINING DETAILS

In this section, we provide the training details of WMPO. The supervised finetuning of OpenVLA-OFT is conducted on 8 H100 GPUs. We exclude robot proprioceptive states and wrist camera inputs for simplicity. Instead of applying L1 regression, we adopt an output head that predicts discrete action tokens in parallel. The subsequent world model training and policy optimization are performed on 32 H100 GPUs.

The world model is first pretrained on the Open X-Embodiment (OXE) dataset Collaboration (2023) and then fine-tuned on downstream policy behavior data. Table 3 summarizes the hyperparameters for training the world model, while the remaining settings follow OpenSora (Zheng et al., 2024). The hyperparameters for policy optimization with GRPO are provided in Tab. 4, with other settings inherited from Li et al. (2025a).

---

**Algorithm 1:** World Model Policy Optimization

**Input** : World model $p_\phi$, reward model $R_\psi$, VLA policy $\pi_\theta$; group size $G$; batch size $B$ (trajectories); mini-batch size $M$ ($M \mid B$); epochs $E$; horizon $T$

**Output** : Updated policy parameters $\theta$

$\theta_{\text{old}} \leftarrow \theta$

**while** *not converged* **do**

  $\mathcal{B} \leftarrow \varnothing$

  **while** $|\mathcal{B}| < B$ **do**

    Sample initial state $s_0 = (I_{0:c}, g) \sim \mathcal{D}$

    **for** $i = 1$ **to** $G$ **do**

      Imagine $\tau_i = \{(s_t^i, a_t^i)\}_{t=0}^T \sim p_\theta(\tau_i \mid s_0, \pi_{\theta_{\text{old}}})$

      $R_i \leftarrow R(\tau_i) \in \{0, 1\}$

    **if** *all*$(R_1, \ldots, R_G)$ *or none*$(R_1, \ldots, R_G)$ **then**

      **continue**

    $\mu \leftarrow \frac{1}{G} \sum_{i=1}^G R_i, \quad \sigma \leftarrow \sqrt{\frac{1}{G} \sum_{i=1}^G (R_i - \mu)^2}$

    **for** $i = 1$ **to** $G$ **do**

      $\hat{A}_i \leftarrow (R_i - \mu)/\sigma$

      Precompute $\{\log \pi_{\theta_{\text{old}}}(a_t^i | s_t^i)\}_{t=0}^T$

      Append $(\tau_i, \hat{A}_i)$ to $\mathcal{B}$

      **if** $|\mathcal{B}| \geq B$ **then**

        **break**

  **for** $e = 1$ **to** $E$ **do**

    **for** $j = 1$ **to** $B/M$ **do**

      $\mathcal{M} \leftarrow j$-th contiguous block of $M$ trajectories from $\mathcal{B}$

      Update $\theta$ according to Eq. 4, where

      $r_{i,t}(\theta) = \exp\big(\log \pi_\theta(a_{i,t}|s_{i,t}) - \log \pi_{\theta_{\text{old}}}(a_{i,t}|s_{i,t})\big)$

  $\theta_{\text{old}} \leftarrow \theta$

---

## B   BASELINE DETAILS

We provide the implementation details of our baselines, including online GRPO and offline DPO. For the GRPO baseline, the main hyperparameters are inherited from Tab. 4. We observe that the batch size has a significant impact on performance. Larger batch sizes (e.g., 64) yield more stable improvements; however, they require a substantial number of real trajectories. Specifically, a single model update requires at least $64 \times 8 = 512$ real trajectories. Moreover, dynamic sampling further filters out groups with a success rate of 0 or 1. As a result, when the rollout budget is $P = 128$, such large batch sizes are infeasible, and even with $P = 1280$, the model can only be updated once or twice. Therefore, we additionally experimented with a smaller batch size of $8$, scaling the

learning rate down proportionally (by a factor of $8$). We report the best results obtained across both configurations (batch size $= 8$ and $= 64$).

For the DPO baseline, we follow the standard preference-based offline training paradigm. Specifically, we construct a preference dataset from trajectories collected by the supervised fine-tuned OpenVLA-OFT model and use it to optimize the policy with the DPO objective. The model architecture and optimizer settings are kept consistent with those used in GRPO for fair comparison. All baselines were trained under the same rollout budgets and evaluation protocols as WMPO to ensure fairness.

## C  REAL WORLD CASES

In this section, we provide additional examples of trajectory predictions made by the world model in real-world settings. Fig 9 illustrates cases where the world model successfully predicts failure trajectories: it has learned that when the square and the stick are misaligned, the square cannot be inserted into the stick. In contrast, Fig 10 shows a failure case where the model does not correctly predict a failed trajectory. Although predictions remain accurate until the final frame, subtle perturbations prevent the model from faithfully capturing the moment when the square gets stuck in the stick. Empirically, such failures are relatively rare on the validation set, indicating that the world model can reliably predict both successful and failed outcomes in the vast majority of scenarios, which is crucial for stable policy optimization.

## D  LIMITATION

While the WMPO framework can in principle support flow-based policies, this work focuses on discretized action representations. As future work, we plan to extend WMPO to more expressive policy classes, such as flow-matching based policies (Black et al., 2024), and explore policy optimization with FlowGRPO (Liu et al., 2025), thereby broadening its applicability across diverse action spaces.

## E  LLM USAGE STATEMENT

GPT-5 was used solely to assist with language refinement and stylistic polishing of the manuscript. The authors confirm that all scientific ideas, study design, data analyses, and conclusions presented in this work are entirely their own. The LLM did not contribute to the generation of research concepts, execution of experiments, or interpretation of findings.

| Hyperparameter | Value |
|---|---|
| Optimizer | AdamW($\beta = 0.9, \beta = 0.999$) |
| Learning rate | 0.0001 |
| Batch size | 128 |
| Gradient clip | 0.1 |
| Pretrain training steps | 12,000,000 |
| Fine-tune training steps | 3,000,000 |
| EMA | 0.9999 |
| Weight decay | 0.0 |
| Prediction target | $\epsilon$ |

Table 3: Hyperparameters for training the world model.

| Hyperparameter | Value |
|---|---|
| Optimizer | AdamW($\beta = 0.9, \beta = 0.999$) |
| Learning rate | $5 \times 10^{-6}$ |
| Training batch size | 64 |
| Group size $G$ | 8 |
| Mini-batch size | 128 |
| Clip ratio $\epsilon_{low}$ | 0.20 |
| Clip ratio $\epsilon_{high}$ | 0.28 |
| Temperature | 1.6 |

Table 4: Hyperparameters for the GRPO algorithm.

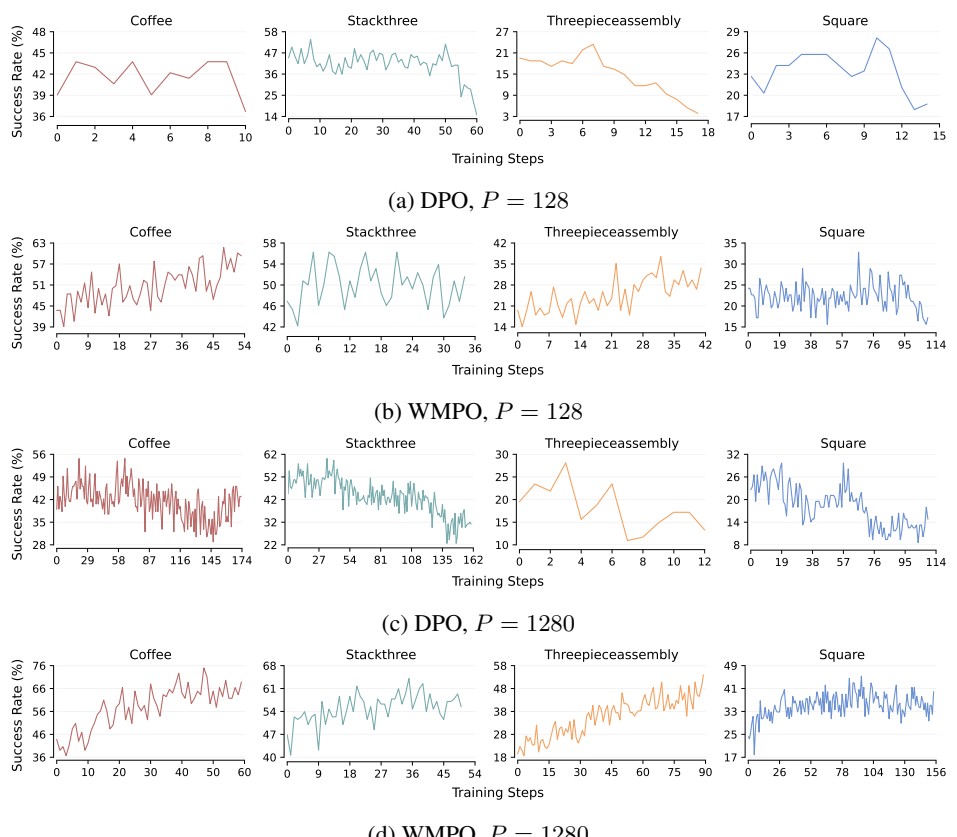

(a) DPO, $P = 128$

(b) WMPO, $P = 128$

(c) DPO, $P = 1280$

(d) WMPO, $P = 1280$

Figure 8: Success rates of DPO and WMPO across four Mimicgen manipulation tasks (Coffee, StackThree, ThreePieceAssembly, and Square) under different rollout budgets ($P = 128$ and $P = 1280$). WMPO consistently achieves more stable improvements and higher final performance compared to DPO, with the performance gap widening under larger rollout budgets.

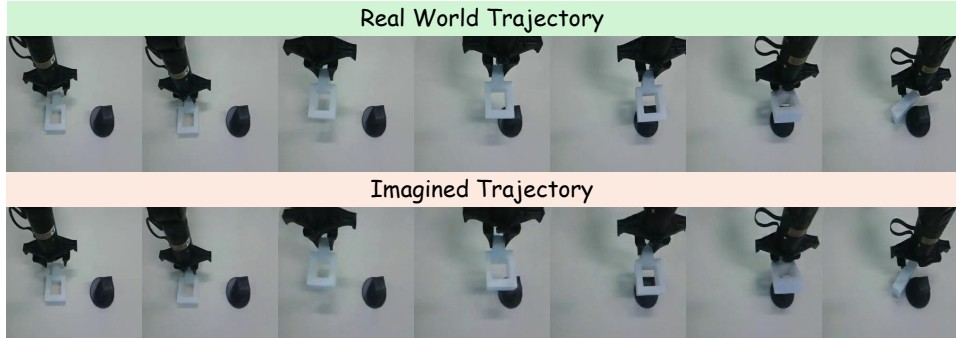

Figure 9: Real-world experiments on the fine-grained manipulation task "Insert the square into the stick". The top row shows the rollout trajectory of the base policy executed in the real world, while the bottom row depicts the corresponding imagined trajectory starting from the same initial state within our learned world model. The world model successfully predicted failure cases.

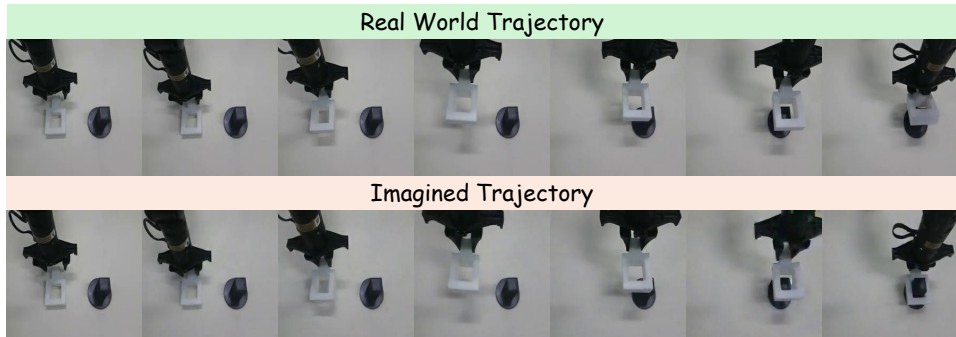

Figure 10: Example of a failure case. Although the predicted trajectory remains accurate until the final frame, the model fails to capture the square getting stuck in the stick due to subtle perturbations.

