# OpenReview forum: "WMPO: World Model-based Policy Optimization for Vision-Language-Action Models"
_ICLR.cc/2026/Conference — ICLR 2026 Poster_

### Official Review · Reviewer_eaSU · 2025-10-24

**Soundness:** 3
**Presentation:** 3
**Contribution:** 3
**Rating:** 6
**Confidence:** 4

**Summary:**

The authors propose a system to train a VLA via RL on trajectories from a world model. A video diffusion model is transformed into a world model by additionally conditioning upon actions. For the reward model, a videoMAE model is finetuned on binary classification on successful/failed trajectories, and the model output probability is used as a reward. The used RL algorithm is GRPO, where the initial frame comes from a real dataset, and the trajectories in the cohort are different sampled videos starting from the same frame. For baseVLA, openvla-OFT is chosen. Across both sim and real, this method beats baselines, and the authors also show that the method improves generalization.

**Strengths:**

- The idea is simple and clearly explained
- The authors show solid improvement on sim and real

**Weaknesses:**

- only a single base-VLA is used (openvla-OFT)
- More real-world experiments would be good
- missing comparison to DreamGen
- the training of the world model lack details

**Questions:**

1. What is meant by 50/1000 steps?
2. Can you compare with PPO as well?
3. Can you try multiple different VLAs?
4. Can you do more real-world experiments?
5. Can you add more details regarding how the world model was trained? Hyperparameters, datasets and so on.

---

> ### Author Response · Authors · 2025-11-26
> **Response to Reviewer eaSU - Thank you for reviewing our paper!**
>
> We sincerely appreciate your time and effort in reviewing our paper! We provide detailed discussions based on your review:
>
> ---
>
> **Q1**: What is meant by 50/1000 steps?
>
> Thank you for the question. In diffusion models, noise is gradually added to data, and usually after **T = 1000** steps the data becomes pure Gaussian noise. To mitigate accumulated error, we inject **50 steps of noise** into the condition frame. This makes the condition frame intentionally imperfect, which effectively reduces the compounding error issue.
>
> ---
>
> **Q2**: Can you compare with PPO as well?
>
> Thank you for the suggestion. Due to time limitations, we may not be able to complete this experiment before the deadline. However, we note that recent work (https://github.com/RLinf/RLinf) has already compared **PPO vs. GRPO** on the **openvla-oft** model, showing that **PPO generally outperforms GRPO**. We plan to include PPO in future experiments.
>
> ---
>
> **Q3**: Can you try multiple different VLAs?
>
> We appreciate the suggestion. We are currently conducting this line of work, and we will update once these experiments are completed.
>
> ---
>
> **Q4**: Can you do more real-world experiments?
>
> Thank you for the suggestion. We have **re-run the real-robot experiments**, due to variations in the real-world experimental setup, increasing the number of trials from **30 to 100**. We additionally evaluated **three generalization settings**:
>
> - Enlarging the random placement range of the square
> - Adding distractor objects in the background
> - Changing the background to blue
>
> The updated results are shown below:
>
> | Setting        | Base Policy | WMPO |
> |----------------|-------------|------|
> | Base setting   | 39          | 58   |
> | Larger range   | 25          | 30   |
> | Distractors    | 22          | 42   |
> | Blue BG        | 22          | 48   |
>
> These results provide **statistically stronger evidence** and demonstrate the **robust generalization** of WMPO in real-world robot tasks.
>
> ---
>
> **Q5**: Can you add more details regarding how the world model was trained? Hyperparameters, datasets, and so on.
>
> Thank you for pointing this out. We provide several training details of the world model in the **Experiment Settings** section, such as the number of conditioning frames and the prediction horizon. Additional hyperparameters and datasets are included in **Appendix A (Training Details)**. For completeness, we summarize the key specifications here.
>
> The training hyperparameters are listed in **Table 3**. The world model is first **pre-trained on the OpenX-Embodiment dataset**, followed by **fine-tuning on policy-behavior rollouts**, collected with a rollout budget of **P = 128 or 1280**. The pre-training and fine-tuning phases take **12 days** and **4 days**, respectively.
>
> We would be happy to provide any further details if needed.

---

### Official Review · Reviewer_vH24 · 2025-10-27

**Soundness:** 2
**Presentation:** 2
**Contribution:** 3
**Rating:** 4
**Confidence:** 2

**Summary:**

The paper proposes World Model-based Policy Optimization (WMPO), a framework for improving Vision-Language-Action (VLA) models via reinforcement learning inside a learned pixel-level video world model. The core idea is to avoid costly real-world robot interactions by performing on-policy RL (specifically GRPO) entirely within an autoregressive video diffusion model pretrained on large-scale robotic data and fine-tuned on downstream policy rollouts. The framework includes: A pixel-space world model for visual fidelity; Policy Behavior Alignment to adapt the world model to policy-induced state/action distributions; A learned binary reward model for task success; Noisy-frame conditioning and frame-level action control to mitigate long-horizon prediction drift; Dynamic Sampling during GRPO to avoid vanishing gradients in sparse-reward settings. Experiments in MimicGen simulation and real-world tasks (e.g., “insert square into stick”) show consistent improvement over imitation learning baselines and offline RL methods such as DPO and limited online GRPO.

**Strengths:**

1. The paper addresses a critical bottleneck in VLA RL: sample inefficiency and brittleness of imitation learning. While prior works (e.g., RT-2, OpenVLA) have shown impressive generalization, they remain confined to IL and struggle to recover from failures. WMPO’s goal — learning to self-correct via on-policy RL in a world model — is both ambitious and well-justified.

2.  The authors’ fine-tuning of the world model on policy-generated trajectories is a principled way to close the distribution shift between expert demonstrations and actual policy rollouts.

3. The qualitative results showing self-correction are compelling. This is not merely improvement in success rate but evidence of learning novel recovery strategies absent in demonstrations.

**Weaknesses:**

1. Overclaiming of “On-Policy” Scalability Without Real Costs: While WMPO avoids real-world rollouts during optimization, it still requires 128–1280 real trajectories to fine-tune the world model and initialize policy behavior alignment. This is not zero-shot or low-data RL — it is offline world-model RL with modest real data.
Recent works like **IRASim** [1] and **World4RL** [2] also use diffusion world models but start from far fewer real trajectories (e.g., 50–100).
The paper does not compare directly to these methods, making it unclear whether the gains come from GRPO or from the specific world-model design.

2. Reward Model is a Black Box with No Robustness Checks. The reward model is trained on binary success labels from real trajectories. But: What if the world model generates semantically correct but visually shifted outcomes (e.g., object slightly displaced)?
There is no evaluation of reward hacking — e.g., does the policy learn to “fool” the reward model by generating plausible but incorrect motions?

3. Comparison to Offline RL is Weak. The DPO baseline is implemented naively: it uses trajectory-level preferences but does not leverage recent advances. Why not compare to IQL + diffusion policy? Also, DPO’s poor performance on background disruption (Tab 2) may reflect overfitting to visual cues, not the inherent weakness of offline RL.

4. The quality of experiments: One task, 30 trials for the real robot task, underpowered for statistical significance. Missing key ablations (e.g., world model w/o policy alignment, latent vs. pixel space, reward model threshold sensitivity).

[1] IRASim: A Fine-Grained World Model for Robot Manipulation

[2] World4RL: Diffusion World Models for Policy Refinement with Reinforcement Learning for Robotic Manipulation

**Questions:**

Please address the aforementioned weakness as thoroughly as possible.

---

> ### Author Response · Authors · 2025-11-26
> **Response to Reviewer vH24 (1/3) - Thank you for reviewing our paper!**
>
> We sincerely appreciate your time and effort in reviewing our paper! We provide detailed discussions based on your review:
>
> ---
>
> **W1**: Overclaiming of “On-Policy” Scalability Without Real Costs: While WMPO avoids real-world rollouts during optimization, it still requires 128–1280 real trajectories to fine-tune the world model and initialize policy behavior alignment. This is not zero-shot or low-data RL — it is offline world-model RL with modest real data. Recent works like IRASim [1] and World4RL [2] also use diffusion world models but start from far fewer real trajectories (e.g., 50–100). The paper does not compare directly to these methods, making it unclear whether the gains come from GRPO or from the specific world-model design.
>
> Thank you for the insightful comments. We address the concerns below.
>
> *1 - Real-World Data Requirements*
>
> Our primary contribution is to demonstrate that **under the same amount of real-world data**, WMPO substantially improves reinforcement learning efficiency by enabling VLA fine-tuning within a learned world model.
> To make this point explicit, we compare WMPO with direct on-policy RL (GRPO) under the same performance target. We measure how many real-environment rollouts GRPO requires to reach the final performance achieved by WMPO. Under this metric, WMPO improves sample efficiency by 10×–50× relative to GRPO.
> In addition, our experiments intentionally evaluate two modest data regimes — **128 and 1280 real trajectories** — to study WMPO’s behavior under different amounts of real data. Across both regimes, WMPO consistently outperforms all baselines.
>
> *2 - Comparison to IRASim and World4RL*
>
> We appreciate the suggestion to discuss IRASim and World4RL. Our rationale for not including direct comparisons is as follows:
>
> - **IRASim** is cited in Section 3.2 (Generative World Model), but it is not designed for VLA RL. Instead, it focuses on policy evaluation and model-based planning, making a direct comparison methodologically misaligned.
>
> - **World4RL** was released on **September 23rd**, one day before the ICLR 2026 submission deadline, leaving insufficient time to incorporate a full comparison. Moreover, World4RL optimizes **Gaussian policies**, whereas our work targets **VLA fine-tuning**, so its setting differs meaningfully from ours.
>
> Regarding data requirements, World4RL does not use “far fewer” real trajectories than WMPO. The method requires:
> - **50 expert trajectories**
> - **150 Gaussian-policy trajectories**
> - **30 random-policy trajectories**
>
> This sums to **230 trajectories**, which is **higher than our low-data setting of 128 trajectories**. Thus, WMPO operates in a comparably or even lower real-data regime.
>
> *3 - On-Policy vs. Offline Terminology*
>
> We agree that WMPO is *offline* in that it does not require online real-environment interaction during optimization. However, because imagined rollouts are always sampled from the *current* policy, WMPO maintains an *on-policy learning signal*. We will revise the manuscript to adjust the terminology accordingly.
>
> We appreciate the reviewer’s suggestions and will incorporate clarifications and a discussion of World4RL in an updated version of the manuscript.

---

> ### Author Response · Authors · 2025-11-26
> **Response to Reviewer vH24 (2/3) - Thank you for reviewing our paper!**
>
> ---
>
> **W2**: Reward Model is a Black Box with No Robustness Checks. The reward model is trained on binary success labels from real trajectories. But: What if the world model generates semantically correct but visually shifted outcomes (e.g., object slightly displaced)? There is no evaluation of reward hacking — e.g., does the policy learn to “fool” the reward model by generating plausible but incorrect motions?
>
> Thank you for the insightful suggestion. We agree that robustness against potential reward-model hacking is an important concern. Below we clarify what is already included in the paper and provide additional details.
>
> *1 - Clarifications on What the Paper Already Addresses*
>
> First, regarding the **reward model**, our framework uses it solely to determine whether a *complete generated trajectory* indicates task success. As reported in the main paper (line 376), the reward model is highly reliable in this setting, achieving an **F1 score above 95%**.
>
> Second, we analyze the failure case of the world model. In **Appendix C** (Real World Cases, see **Figure 9**), we explicitly discuss cases where the world model produces *plausible but incorrect* imagined outcomes. For example, the real robot trajectory results in the block getting stuck, whereas the world model incorrectly predicts a successful placement. These failure modes occur but are relatively few on our validation set.
>
> ---
>
> *2 - A More Detailed Breakdown of Failure Modes*
>
> To more precisely characterize the robustness issue raised in the question, we distinguish between two different error sources:
>
> 1. **World-model prediction errors**
>    These occur when the **same full trajectory** leads to different outcomes in the real world versus the world model’s imagined rollout.
>
> 2. **Reward-model prediction errors**
>    These occur when the reward model **misclassifies** an imagined trajectory—e.g., granting success to trajectories that should be failures (or vice versa).
>
> This separation helps us understand whether errors arise from the world model or from the reward model itself.
>
> From our real-world experiments, we observe that:
> - **World-model prediction errors** occur in fewer than **15%** of cases.
> - **Reward-model prediction errors** occur in fewer than **2%** of cases.
>
> Although these imperfections are inevitable in any world-model-based VLA RL system, our results show that **WMPO still consistently improves task performance across all tasks**. This suggests that the advantages of world-model policy optimization **significantly outweigh** the occasional issues arising from world-model hallucination or reward-model misclassification.
>
> *3 - Addressing the Concern About Reward Hacking*
>
> "Reward hacking" refers to a policy intentionally exploiting the reward system by generating trajectories that score as successful without actually accomplishing the task. In WMPO, this is structurally difficult: the reward model only labels imagined trajectories, while the success rates in Table 1 are determined by the simulator, which provides ground-truth outcomes and is not influenced by the learned reward model.
>
> To check explicitly, we manually inspected imagined-success cases. We found no systematic pattern suggesting that the policy attempts to "fool" the reward model (e.g., through subtly incorrect but visually plausible motions). All mismatches are attributable to ordinary world-model prediction noise rather than adversarial exploitation.
>
> Overall, although neither the world model nor reward model is perfect, we observe no evidence of reward hacking, and WMPO remains effective in real-world execution.
>
> We appreciate the reviewer’s suggestion and will revise the presentation to make this robustness analysis clearer. As both world models and robot reward models continue to improve, the effectiveness of WMPO—as a general, model-agnostic world-model-based RL framework—will naturally continue to increase.

---

> ### Author Response · Authors · 2025-11-26
> **Response to Reviewer vH24 (3/3) - Thank you for reviewing our paper!**
>
> ---
>
> **W3**: Comparison to Offline RL is Weak. The DPO baseline is implemented naively: it uses trajectory-level preferences but does not leverage recent advances. Why not compare to IQL + diffusion policy? Also, DPO’s poor performance on background disruption (Tab 2) may reflect overfitting to visual cues, not the inherent weakness of offline RL.
>
> Thank you for the thoughtful suggestion. We follow the **standard DPO implementation**, and our DPO baseline demonstrates clear effectiveness in **Table 1**, **Figure 6**, and **real-world experiments**.
>
> Regarding comparisons to **IQL + diffusion policy**, our primary reason for not including this baseline is that **WMPO is designed to optimize VLA models**. While we agree that IQL is a promising offline RL algorithm, there is **currently no mature and stable solution** for applying IQL to VLA models in an end-to-end RL setting. In contrast, DPO is a **well-established and practically usable** baseline for VLA RL[1][2], which makes it a more appropriate comparison in our current framework. Exploring **IQL with VLA models** is an exciting direction, and we consider it **valuable future work**.
>
> We also agree that the poor performance of DPO under **background disruption (Table 2)** reflects the **limitations of our current DPO implementation**, rather than an inherent weakness of offline RL itself.
>
> We appreciate your insightful feedback and will update the manuscript to clarify these points.
>
> [1] GRAPE: Generalizing Robot Policy via Preference Alignment. arXiv:2411.19309. 2014.
> [2] Human-assisted Robotic Policy Refinement via Action Preference Optimization. NeurlPS. 2025.
>
> ---
>
> **W4**: The quality of experiments: One task, 30 trials for the real robot task, underpowered for statistical significance. Missing key ablations (e.g., world model w/o policy alignment, latent vs. pixel space, reward model threshold sensitivity).
>
>
> Thank you for the valuable suggestion. We address the comment in two parts.
>
>
> *1. Real-World Experiment Scale and Statistical Significance*
>
> To strengthen the statistical validity of our real-robot evaluation, we **increased the number of trials from 30 to 100** and additionally evaluated the policy under **three generalization settings**:
>
> - Larger random placement range
> - Distractor objects in the background
> - Blue background
>
> The updated success counts are:
>
> | Setting      | Base Policy | WMPO |
> |--------------|-------------|------|
> | Base setting | 39          | 58   |
> | Larger range | 25          | 30   |
> | Distractors  | 22          | 42   |
> | Blue BG      | 22          | 48   |
>
> These expanded results offer **much stronger statistical confidence** and show that WMPO delivers **consistent and robust real-world improvements** across diverse conditions.
>
>
> *2. Missing Ablations*
>
> **(a) World model without policy alignment.**
> As noted around *line 247*, an unaligned world model almost never predicts failure trajectories correctly, producing rollouts that are overwhelmingly “successful.” This leads to **uninformative reward signals**, causing the RL process to fail directly.
>
> **(b) Pixel space vs. latent space dynamics.**
> As discussed in *lines 046–072*, we intentionally model dynamics in **pixel space**, the unified observation space for VLA models. Latent-space world models are difficult to use here because **different VLA architectures produce incompatible latent representations**, preventing consistent RL training or scalability.
>
> **(c) Reward-model threshold sensitivity.**
> The threshold is selected via grid search on a validation set and achieves **high F1 scores** (>95%). Empirically, the performacee of reward model is **stable across a reasonable range of threshold values**, indicating low hyperparameter sensitivity.
>
> In summary, we hope that the expanded real-world evaluation and the additional ablations address your concerns and further strengthen the empirical support for WMPO.

---

### Official Review · Reviewer_Wgig · 2025-10-29

**Soundness:** 4
**Presentation:** 4
**Contribution:** 4
**Rating:** 6
**Confidence:** 3

**Summary:**

This paper proposes WMPO (World-Model-based Policy Optimization), an on-policy RL framework for VLA models that replaces costly real-robot rollouts with trajectories “imagined” by a pixel-space video world model. Key ingredients are: (1) a diffusion video world model with noisy-frame conditioning for robustness and frame-level action control for precise action–frame alignment; (2) policy behavior alignment, i.e., fine-tuning the world model on the policy’s own rollouts to match failure as well as success states; (3) a lightweight clip-based reward model for outcome (0/1) labeling; and (4) on-policy GRPO training with dynamic sampling and no KL term (no reference model). The method aims to keep the visual state space aligned with pretrained VLA encoders by decoding back to pixels. Experiments on Mimicgen show consistent gains over GRPO and DPO baselines at two real-rollout budgets (e.g., mean SR 47.1% vs. 37.3% at P=128 and 57.6% vs. 42.4% at P=1280). WMPO also reports emergent self-correction, improved generalization under spatial/background/texture shifts, lifelong learning improvements via alternating policy/world-model updates, and a real-robot result on “Insert the square into the stick”

**Strengths:**

Clear, modular recipe: world-model rollouts + outcome classifier + GRPO, with practical choices (pixel-space decoding to match VLA features; noisy-frame conditioning; frame-level action injection). The write-up is concrete and reproducible.


Consistent empirical gains over strong baselines across four Mimicgen tasks and two rollout budgets; improvements grow with budget (data-efficiency + scaling).


Behavioral insights: convincing qualitative evidence of self-correction and reduced “getting stuck,” with trajectory-length analysis.


Generalization: better robustness under spatial/background/texture disruptions than baselines.

**Weaknesses:**

Heavy compute / practicality: training uses 32× H100 for world-model/WMPO phases (plus 8× H100 for SFT). The paper would benefit from wall-clock, throughput, and ablations on smaller budgets/hardware.

Model-world fidelity & safety: while qualitative results are strong, there’s limited quantitative assessment of rollout fidelity (e.g., per-step action-conditioned metrics), failure taxonomy, or safety constraints—especially since outcome-only rewards can reward shortcuts.

**Questions:**

1. please try to handle these weaknesses

2. Do you try to use some PeFT methods for low-resource environments? Maybe these discussions will help

---

> ### Author Response · Authors · 2025-11-26
> **Response to Reviewer Wgig (1/2) - Thank you for reviewing our paper!**
>
> We sincerely appreciate your time and effort in reviewing our paper! We provide detailed discussions based on your review:
>
> ---
>
> **W1**: Heavy compute / practicality: training uses 32× H100 for world-model/WMPO phases (plus 8× H100 for SFT). The paper would benefit from wall-clock, throughput, and ablations on smaller budgets/hardware.
>
> Thank you for raising this important point. Our detailed responses are organized as follows.
>
> *1 - Wall-clock & throughput.*
>
> For **world-model training**, Table 3 in the paper specifies all hyperparameters.
> Finetuning the world model requires **~4 days for 300k steps on 32× H100 GPUs**.
>
> For **policy optimization**, each task is trained to near convergence.
> Below are the training steps and wall-clock hours:
>
> | Task | Coffee | StackThree | ThreePieceAssembly | Square |
> |------|--------|------------|--------------------|--------|
> | **Rollout budget P = 128** |||||
> | Training steps | 48 | 21 | 32 | 67 |
> | Hours | 38 | 16 | 29 | 32 |
> | **Rollout budget P = 1280** |||||
> | Training steps | 47 | 35 | 89 | 94 |
> | Hours | 24 | 26 | 79 | 39 |
>
>
> *2 - Trade-off Between Computational Resources and Real-World Data*
>
> Our core observation is that **WMPO improves RL efficiency by scaling compute**, not by requiring more real-world data.
> Under **fixed real-world data**, increased compute for world-model training and rollouts significantly enhances learning efficiency.
>
> This represents a **complementary scaling axis** for robotics:
> - Traditional axis: collect more real-world experience
> - WMPO axis: increase compute to extract more value from the same real data
>
> WMPO is thus **compatible with additional real-world data** when available, and can leverage it more efficiently via world-model-based RL.
>
> *3 - Sample Efficiency Compared to Direct RL (GRPO)*
>
> To highlight the benefit of this compute-based scaling, we compare WMPO to direct RL using GRPO in terms of **real-world rollouts required to reach the same final performance**.
>
> Across tasks, **WMPO achieves 10×–50× improvements in sample efficiency**, demonstrating that compute can effectively substitute for large amounts of real data.
>
> ####  Sample-Efficiency Improvements
>
> | Task                   | WMPO vs. GRPO Improvement |
> |------------------------|----------------------------|
> | Coffee                 | **50×**                    |
> | StackThree             | **20×**                    |
> | ThreePieceAssembly     | **50×**                    |
> | Square                 | **10×**                    |
>
> *4 - Ablations on smaller budgets.*
>
> Although our main experiments use **32× H100 GPUs**, this setup primarily accelerates:
> - world-model training
> - imagined rollout generation
>
> Importantly, **WMPO does not intrinsically require large-scale hardware**.
> To evaluate WMPO under constrained compute, we perform policy optimization phrase using **8× H100 GPUs**. WMPO maintains strong performance with only slight variance due to distributed numerical effects.
>
> | Setting | Method      | StackThree | Square |
> |---------|-------------|------------|--------|
> | -  | Base Policy | 46.9 | 24.2 |
> | 32 GPUs | WMPO        | 64.1 | 45.3 |
> | 8 GPUs  | WMPO        | 64.1 | 44.5 |
>
> WMPO consistently yields substantial gains even on reduced compute, with the only cost being longer training time.

---

> ### Author Response · Authors · 2025-11-26
> **Response to Reviewer Wgig (2/2) - Thank you for reviewing our paper!**
>
> ---
>
> **W2**: Model-world fidelity & safety: while qualitative results are strong, there’s limited quantitative assessment of rollout fidelity (e.g., per-step action-conditioned metrics), failure taxonomy, or safety constraints—especially since outcome-only rewards can reward shortcuts.
>
> We thank the reviewer for this insightful comment.
> We evaluated WMPO in the real world and additionally tested its performance on the **quantitative assessment of rollout fidelity**. Specifically, we compared the PSNR between the video predicted from the full trajectory and the given first frame, and the corresponding real video. The resulting PSNR is **21.3 dB**, which is a promising result. As a reference, in [1], Cosmos[1] and IRASim[2] report PSNR scores of **21.1 dB** and **19.1 dB**, respectively, on the Bridge dataset. This indicates that our world model is capable of generating **high-fidelity predictions**, which forms the basis for strong qualitative performance in RL.
>
> In Figure 9, we visualize an example of a world-model prediction failure, where the real trajectory results in the square getting stuck, whereas the world model predicts that the square can be placed correctly. In fact, the failure taxonomy can be divided into two categories:
>
> 1. **World-model prediction errors**, where the same full action-trajectory leads to different outcomes in the real environment versus the world model.
> 2. **Reward-model prediction errors**, where the reward model misclassifies the imagined trajectories.
>
> In the real-world setting, we found that the first type of error occurs in **fewer than 15%** of cases, while the second type occurs in **fewer than 2%** of cases. Although both types of errors are unavoidable under the world-model-based VLA RL framework, we observe that the model is still able to improve performance despite them. This suggests that the gains brought by WMPO **significantly outweigh** issues caused by world-model hallucination and reward-model hacking.
>
> Future advances in world models and reward models will further mitigate these issues, and WMPO—being a general world-model-based VLA RL framework—will likely become even more effective under improved model fidelity and robot reward model.
>
> ---
>
> **Q2**: Do you try to use some PeFT methods for low-resource environments? Maybe these discussions will help.
>
> We appreciate the suggestion to explore PeFT. Following this advice, we applied LoRA to VLA RL training within WMPO. However, this resulted in unstable optimization: under identical settings, full-parameter fine-tuning consistently improved performance, whereas LoRA struggled to produce meaningful gains.
>
> This observation is consistent with prior findings:
> - Qwen2 exhibits degraded math-reasoning performance when LoRA is applied during RL optimization [3].
> - RL with LoRA in LLMs is known to be non-trivial, often requiring careful tuning of multiple hyperparameters to ensure stability [4] (see Section “Setting LoRA hyperparameters”).
>
> Since there is currently **no effective LoRA-based RL method for VLAs**, we leave LoRA-based VLA RL fine-tuning as an important direction for future research.
>
> We also note that **rollout generation dominates the overall training time**, due to the need to synthesize large quantities of videos. To mitigate this, we opted to allocate additional computational resources to accelerate the rollout process. Future work may incorporate **diffusion distillation** (e.g., Shortcut Models [5]) to further reduce rollout costs and improve applicability in low-resource settings.
>
> ---
>
> [1] Cosmos World Foundation Model Platform for Physical AI. arxiv:2501.03575
> [2] IRASim: A Fine-Grained World Model for Robot Manipulation, ICCV 2025.
> [3] Token-Efficient RL for LLM Reasoning. arXiv:2504.20834
> [4] https://thinkingmachines.ai/blog/lora
> [5] One Step Diffusion via Shortcut Models. arXiv:2410.12557

---

> > ### Comment · Reviewer_Wgig · 2025-11-26
> >
> > Dear Authors,
> > Thank you for your insightful rebuttal. I update my score to 8. Hope your paper will be accepted.

---

> > > ### Author Response · Authors · 2025-11-26
> > >
> > > Thank you for your kind support and encouraging score update. We sincerely appreciate your recognition of our work!

---

### Official Review · Reviewer_VLDi · 2025-10-30

**Soundness:** 3
**Presentation:** 2
**Contribution:** 2
**Rating:** 4
**Confidence:** 3

**Summary:**

WMPO proposes a novel policy finetuning method without using online interaction by using a world model on the pixel space, which allows robust finetuning of vision-language-action models on imagined trajectories. There are a few desirable properties of this method compared to traditional world modeling and finetuning objectives:
1. The world model can directly be trained in pixel space, which allows a pixel based policy to interface with the world model without extra decoders.
2. The method can be completely open sourced, and can be adapted into other distributions.
3. This method allows on policy finetuning of VLAs using GRPO without any real-world demonstrations or preexisting datasets.
The authors then used GRPO to perform finetuning of a base OpenVLA-OFT policy, which shows desirable success rate, scalability, and robustness in RoboMimic. On the real world, WMPO also shows better performance compared to other finetuning methods such as direct GRPO and DPO.

**Strengths:**

1. The method section of the paper is concise and informative.
2. I believe that there are sufficient ablations being done on the method, and I like that the authors have demonstrated good robustness of the method when dealing with OOD settings and desirable scalability.

**Weaknesses:**

1. I believe that the paper did not address how this method can be extended into a generalist setting. The scope of the environment is also rather limited, albeit there are adequate ablations being conducted.
2. In addition, the paper used OpenVLA-OFT as the base policy. This again limits how much promise the method can bring to generalist policies. If the authors can provide additional ablations without using OpenVLA-OFT, I believe this can strengthen the paper.
3. I believe that the paper did not address the question of how adaptable this method is when concerned with language-conditioned settings (even though OpenVLA is language conditioned, there are no mention of how to use language labels in this paper), furthering this discussion can also be beneficial to the rating of the paper.

**Questions:**

1. I might have missed this, but when you are increasing the rollout budget, do you use the larger set as well to finetune the world model?
2. One reason for not using real-world demonstrations at all is due to it being more expensive. Have you considered using a few real-world demonstrations as regularization in your method, and if so, how does it compare to only using imagined trajectories?
3. It seems that the authors chose to use on policy RL because of flaws with off policy RL [1] and propagating the correct value. Are there going to be any potential concerns when implementing such a method in the off-policy setting?

Minor remarks:
1. The main figure stated that BC from human demonstrations cannot achieve self-corrective behavior, but this is not correct when the training data contains corrective behavior. Similarly, you can do on-policy RL in real world if you can run it online.
2. Follow up on q3, it would be better to show differences in scalability if you have to use an off policy setting.

References:
[1] Park, S. et al., 2025. “Horizon Reduction Makes RL Scalable.” NeurIPS.

---

> ### Author Response · Authors · 2025-11-26
> **Response to Reviewer VLDi (1/2) - Thank you for reviewing our paper!**
>
> We sincerely appreciate your time and effort in reviewing our paper! Following your feedback, we conducted additional experiments and we hope these new results and discussions will effectively address your concerns:
>
> ---
>
> **W1**: I believe that the paper did not address how this method can be extended into a generalist setting. The scope of the environment is also rather limited, albeit there are adequate ablations being conducted.
>
> Thank you for the constructive feedback.
>
> *1 - WMPO in a generalist setting*
>
> Our formulation of WMPO in **Section 3.1 (Problem Formulation)** is inherently generalist: the framework itself is task-agnostic and supports generalist setting. The Mimicgen benchmark used in our main experiments was chosen because its tasks require fine-grained manipulation skills that are difficult to acquire through imitation alone and therefore provide a strong testbed for comparing RL algorithms.
>
>
> To address this, we conducted additional experiments on the LIBERO benchmark. The libero-goal suite contains 10 heterogeneous manipulation tasks across kitchen, living-room, and study environments, covering multi-object placement, container interaction, appliance operation, and standard pick-and-place tasks. These tasks serve as a reasonable proxy for assessing generalist manipulation behavior.
>
> *2 -  Evaluation in a broader context*
>
> Moreover, we agree that evaluating WMPO in a broader generalist context is important. To address this, we conducted additional experiments on the LIBERO benchmark. The libero-goal suite contains 10 heterogeneous manipulation tasks across kitchen, living-room, and study environments, covering multi-object placement, container interaction, appliance operation, and standard pick-and-place tasks. These tasks serve as a reasonable proxy for assessing generalist manipulation behavior.
> For the experiment setup, we first use a pretrained VLA (OpenVLA-OFT) to collect **1,000 policy-behavior trajectories per task** (10,000 in total). We then fine-tune our world model on these real trajectories and train the VLA within the learned world model. As a baseline, we include **GRPO**, which performs direct RL in the simulator using the same number of real trajectories (1,000 per task).
> As shown in the following table, WMPO significantly outperforms GRPO under the same real-world rollout budget:
>
>
> | Method   | Success Rate (%) |
> |----------|------------------|
> | Base Policy | 42.4             |
> | GRPO     | 46.6             |
> | **WMPO (ours)** | **55.0**     |
>
> We attribute this improvement in sample efficiency to the ability of our method to leverage imagined trajectories generated by the world model for continued policy optimization, whereas direct RL relies solely on real rollouts.
>
> These additional results demonstrate that WMPO applies to a generalist manipulation setting without any modification and that a fine-tuned world model provides a strong learning scaffold for efficient VLA improvement. We appreciate the reviewer’s suggestion and will incorporate these findings and clarifications into the revised version of the paper.
>
> ---
>
> **W3**: I believe that the paper did not address the question of how adaptable this method is when concerned with language-conditioned settings (even though OpenVLA is language conditioned, there are no mention of how to use language labels in this paper), furthering this discussion can also be beneficial to the rating of the paper.
>
> Thank you for the helpful suggestion. We agree that clarifying the language-conditioned aspect of our method can strengthen the paper. As described in **Section 3.1 (Problem Formulation)**, the state space in our framework explicitly includes the language instruction, and **Figure 2** (upper left) provides an example instruction (“Insert the square into the stick”).
>
> In WMPO, we follow the standard language-conditioned setup used in prior VLA frameworks and feed the language instruction into the VLA during policy optimization. We do not provide language instructions to the world model, as the world model is only required to generate trajectories conditioned on the policy’s actions.
>
> Moreover, our additional experiments on the **LIBERO** benchmark demonstrate that WMPO can effectively optimize a **language-conditioned multi-task VLA**. These results provide empirical evidence that our method is effective in language-conditioned settings.

---

> ### Author Response · Authors · 2025-11-26
> **Response to Reviewer VLDi (2/2) - Thank you for reviewing our paper!**
>
> ---
> **Q1**: I might have missed this, but when you are increasing the rollout budget, do you use the larger set as well to finetune the world model?
>
> Yes. The core objective of our paper is to investigate whether incorporating a world model can improve the sample efficiency of VLA RL. Since we introduce a real rollout budget that limits the amount of real trajectory available, our comparisons focus on how effectively different RL methods can utilize this constrained budget.
>
> Therefore, when a larger rollout budget is provided, we naturally obtain more policy behavior data (larger set). In such cases, we use the data to fine-tune the world model, allowing it to better approximate the real environment and generate more reliable imagined rollouts for policy optimization.
>
> ---
>
> **Q2**: One reason for not using real-world demonstrations at all is due to it being more expensive. Have you considered using a few real-world demonstrations as regularization in your method, and if so, how does it compare to only using imagined trajectories?
>
> Thank you for the thoughtful suggestion. WMPO naturally supports improvements about the RL algorithms. To verify this, we conducted an experiment in which we incorporated real expert trajectories as a regularization signal. Specifically, we sampled real expert trajectories with the same batch size as imagined trajectories and trained the model using a mixture of SFT loss and RL loss. In general, we set the SFT loss weight to **0.005**, under which the **value of the RL loss is roughly 1.6× larger** than the SFT loss during training. After smoothing the success rate curves to reduce randomness, the results on the **Coffee** task are shown below:
>
> | Train Step                 | 0      | 10     | 20     | 30     | 40     |
> |----------------------------|--------|--------|--------|--------|--------|
> | WMPO with real trajectory  | 43.75  | 47.8   | 52.5   | 56.8   | 60.2   |
> | WMPO                       | 43.75  | 43.2   | 52.4   | 57.4   | 63.0   |
>
> We observe that using real expert trajectories as regularization can accelerate early performance gains, but it ultimately **limits the final performance**. Further analysis shows that the RL loss remains stable around **0.005**, while the SFT loss gradually increases from **0.001 to 0.005**. This suggests that the VLA progressively diverges from the initial SFT model; once the SFT loss becomes comparable to the RL loss, it begins to **constrain the model’s improvement**, which explains the reduced final performance.
>
> ---
>
> **Q3**: It seems that the authors chose to use on policy RL because of flaws with off policy RL [1] and propagating the correct value. Are there going to be any potential concerns when implementing such a method in the off-policy setting?
>
> We thank the reviewer for the insightful suggestion. We would like to clarify that our framework is agnostic to the choice of RL algorithms. The core contribution of WMPO is a framework for world-model-based RL that helps perform real-robot RL without actually interacting with the environment. Given this framework, we can easily swap the underlying RL algorithms, including off-policy algorithms. The reason we choose not to, is the key benefit of world-model-based RL is exactly to enable highly-performant on-policy RL [2,3,4,5,6,7], as we can perform parallel rollouts. However, we also agree that further investigations into off-policy settings could be helpful as in the future, given large foundation world models, performing large-scale rollouts could be expensive in cost-sensitive settings. We leave it for future study. We thank the reviewer again and will improve our presentation in the final version.
>
> ---
>
> **W2**: In addition, the paper used OpenVLA-OFT as the base policy. This again limits how much promise the method can bring to generalist policies. If the authors can provide additional ablations without using OpenVLA-OFT, I believe this can strengthen the paper.
>
> Thank you for the constructive suggestion.
> We are currently conducting experiments using a broader variety of VLAs as the base policy, and we will update the results once these experiments are completed.
>
> [1] Horizon Reduction Makes RL Scalable. NeurIPS. 2025.
> [2] Dream to Control: Learning Behaviors by Latent Imagination. arXiv:1912.01603. 2019.
> [3] Mastering Atari with Discrete World Models. arXiv:2010.02193. 2020.
> [4] Mastering diverse control tasks through world models. Nature. 2025.
> [5] Temporal Difference Learning for Model Predictive Control. arXiv:2203.04955. 2022.
> [6] TD-MPC2: Scalable, Robust World Models for Continuous Control. arXiv:2310.16828v2. 2023.
> [7] “DayDreamer: World Models for Physical Robot Learning. CoRL. 2022.

---

### Author Response · Authors · 2025-12-03
**General Response**

Dear Area Chair and Reviewers,

We sincerely thank the reviewers for their thoughtful feedback and constructive suggestions. We are particularly encouraged that **Reviewer Wgig has raised their score to 8 and explicitly supports the acceptance of our paper**, recognizing our informative rebuttal and the strengths of our work. Additionally, we strictly followed **Reviewer VLDi’s suggestion** that addressing adaptability in language-conditioned settings would be **"beneficial to the rating of the paper"**, and have thoroughly addressed this through extensive new experiments on the language-conditioned LIBERO benchmark.

Below we summarize the key strengths recognized by the reviewers and the major improvements made during the rebuttal:

## Review Highlights
* **Motivation:** Addresses a critical bottleneck in VLA RL: sample efficiency (Wgig, vH24).
* **Method:** Simple and informative (VLDi, eaSU) with a clear, modular recipe (Wgig).
* **Experiments:** Solid improvement on sim and real (eaSU), desirable scalability (VLDi, Wgig), generalization (VLDi, Wgig), behavioral insights (Wgig, vH24) and sufficient ablations (VLDi).

## Summary of Rebuttal Updates
Since the initial review, we have actively engaged with all reviewers and addressed their concerns through extensive additional experiments. Below, we summarize the major updates and additional evaluations conducted during the rebuttal period.

**1. Extensive Real-World Evaluation (Addressing vH24, eaSU, VLDi).**
To address concerns regarding statistical significance (previously 30 trials) and limited environmental scope, we significantly expanded our real-robot evaluation:
* **Increased Sample Size:** We increased the number of evaluation trials from **30 to 100**.
* **Robustness Testing:** We evaluated the policy under three challenging generalization settings: (1) Larger random placement range, (2) Distractor objects, and (3) Unseen background colors.
* **Results:** WMPO demonstrated strong robustness, achieving **58/100** successes (vs. **39/100** for Base Policy) in the standard setting, and maintaining significant margins across all generalization settings.

**2. Generalization to Diverse Tasks & Language Conditioning (Addressing VLDi).**
To address the concern regarding generalist capabilities and language conditioning, we conducted new experiments on the **LIBERO-Goal benchmark**, which contains 10 heterogeneous, language-conditioned manipulation tasks.
* **Results:** WMPO achieved a **55.0%** success rate compared to **46.6%** for GRPO and **42.4%** for the base policy under the same real-world rollout budget. This confirms that WMPO applies effectively to generalist, language-conditioned settings without modification.

**3. Computational Analysis & Ablations (Addressing Wgig, VLDi).**
* **Resource Constraints:** We performed an ablation using **8× H100 GPUs** (down from 32×) for policy optimization, showing that WMPO maintains strong performance (**64.1%** Success Rate) even with reduced compute resources.
* **LoRA Exploration:** As suggested by Reviewer Wgig, we investigated Low-Rank Adaptation (LoRA). Our results indicate that LoRA is currently unstable for VLA RL compared to full fine-tuning, highlighting an important direction for future research.
* **Real-Data Regularization:** As suggested by Reviewer VLDi, we tested adding real-world demonstrations as a regularization term. Results showed that while this accelerates early training, it ultimately limits final performance compared to original WMPO, validating our design choices.

**4. Model Fidelity & Failure Analysis (Addressing vH24).**
We provided a detailed analysis of world model fidelity and potential reward hacking:
* **Quantitative:** Our world model achieves a PSNR of **21.3 dB**, competitive with state-of-the-art models like Cosmos and IRASim.
* **Failure Taxonomy:** We found that world-model prediction errors occur in <15% of cases, and reward-model errors in <2% of cases. Despite these imperfections, the RL signal remains sufficiently strong to drive consistent policy improvement, and we observed no evidence of "reward hacking" in the simulation, indicating that our model learns genuine manipulation skills rather than exploiting simulator loopholes.

We believe these additional experiments and clarifications substantially strengthen the paper and thoroughly address the reviewers' concerns. We will incorporate all these new findings into the final version of the paper. We thank the reviewers again for their time and valuable insights.

Sincerely,

The Authors

---

### Meta-Review · Area_Chair_xXjC · 2026-01-07

**Summary:**

WMPO proposes a world-model-based framework for optimizing VLA policies via on-policy reinforcement learning conducted within an action-conditioned, pixel-space video world model, enabling imagined rollouts that remain aligned with pretrained VLA visual features. Reviewers' initial concerns focused on computational practicality and missing wall-clock analysis, world-model fidelity and potential reward-hacking or safety issues, overstatement of on-policy scalability without real-world cost, and limited evaluation across base models and task settings. After rebuttal, many concerns have been adequately addressed, and reviewers are largely positive; I therefore recommend acceptance.

**Reviewer Concerns:**

Concerns largely addressed by the rebuttal
- [VLDi] limited evaluations; Language-conditioning; Use of real-world demonstrations as regularization; Rollout budget
- [Wgig] Heavy compute; fidelity metrics; error rate; LoRA experiments
- [vH24] overclaim on on-policy scalability:; metrics & error rate; Real-robot experiment scale; Missing ablations
- [eaSU] Real-world evaluation scale; Clarification of diffusion steps & world model training details

Concerns still outstanding / partially answered;
- [VLDi] OpenVLA-OFT as the only base policy; Off-policy setting
- [vH24] comparisons to recent works; comparison to offline RL
- [eaSU] OpenVLA-OFT as the only base policy; comparison to PPO; DreamGen

**Reviewer Scores:**

- [VLDi] likely keep 4 or raise to 6
- [Wgig] likely keep 6 or raise to 8
- [vH24] likely remain at 4
- [eaSU] likely keep 6

---

### Decision · Program_Chairs · 2026-01-26

Accept (Poster)